# *Saccharomyces cerevisiae*, a Powerful Model for Studying rRNA Modifications and Their Effects on Translation Fidelity

**DOI:** 10.3390/ijms22147419

**Published:** 2021-07-10

**Authors:** Agnès Baudin-Baillieu, Olivier Namy

**Affiliations:** Institute for Integrative Biology of the Cell (I2BC), Université Paris-Saclay, CEA, CNRS, 91198 Gif-sur-Yvette, France; agnes.baudin-baillieu@i2bc.paris-saclay.fr

**Keywords:** RNA modifications, ribosomes, translation fidelity, *Saccharomyces cerevisiae*

## Abstract

Ribosomal RNA is a major component of the ribosome. This RNA plays a crucial role in ribosome functioning by ensuring the formation of the peptide bond between amino acids and the accurate decoding of the genetic code. The rRNA carries many chemical modifications that participate in its maturation, the formation of the ribosome and its functioning. In this review, we present the different modifications and how they are deposited on the rRNA. We also describe the most recent results showing that the modified positions are not 100% modified, which creates a heterogeneous population of ribosomes. This gave rise to the concept of specialized ribosomes that we discuss. The knowledge accumulated in the yeast *Saccharomyces cerevisiae* is very helpful to better understand the role of rRNA modifications in humans, especially in ribosomopathies.

## 1. Introduction

The ribosome is a complex system that translates the nucleotide code of messenger RNA into protein in cells. Eukaryotic ribosomes consist of four RNA species and more than 80 proteins, forming two independent subunits—the large subunit (LSU) and the small subunit (SSU)—which are connected when mRNA is recruited, adopting a conformation that delimits regions in which base pairing can occur between tRNA and mRNA, leading to the synthesis of a nascent polypeptide chain. In eukaryotes, the RNA moiety—consisting of the 25S, 5.8S and 5S rRNAs for the LSU and the 18S rRNA for the SSU, in yeast—has long been known to contain two regions essential for key catalytic activities for translation: the decoding center (DC) and the peptidyl transferase center (PTC). These rRNAs are transcribed as a long 35S primary transcript encompassing the 18S, 5.8S and 25S mature rRNAs, the 5S rRNA being transcribed independently. Maturation steps occur in parallel with rRNA folding and ribosomal protein assembly and are driven by more than 200 *trans*-acting partners (for review see [1]). During these complex processing events, rRNA undergoes many posttranscriptional modifications. In total, 112 positions have been shown to be modified in the yeast *Saccharomyces cerevisiae,* with 12 different classes of modified nucleotides based on simple chemical transformations and a single, complex, multistep modification. Uridine-to-pseudouridine isomerization and methylation at the ribose 2′OH predominate. Both these modifications are mediated by RNP complexes consisting of an enzyme catalyzing the modification itself and a guide RNA, which identifies the nucleotide targeted. In this review, we will focus on rRNA modifications, explaining how they occur, their mode of action and their functional consequences for ribosome function. The ribosome was long thought to have a very stable structure, but it has emerged in recent years that, on the contrary, it is highly flexible in terms of its composition and modifications. Our vision of the ribosome has, thus, evolved considerably, from a ribosome with a fixed composition reading the genetic code linearly, to a ribosome with a dynamic composition playing an integral role in the regulation of gene expression. This is how the notion of specialized ribosomes emerged, according to which some mRNAs are translated by ribosomes with a specific composition. The rRNA biogenesis and modification pathways in the yeast *Saccharomyces cerevisiae* are a fantastic tool, providing us with essential knowledge to underpin future studies in more complex organisms.

## 2. RNA-Guided Modifications

The vast majority of rRNA modifications are guided by two classes of small nucleolar RNAs (snoRNAs)—box H/ACA and C/D snoRNAs—in a process based on mechanisms that have been extensively studied. In the late 1980s and early 1990s, a large number of snoRNAs were identified on the basis of three criteria: (i) nucleolar localization, (ii) association with rRNA or nucleolar protein (Nop1) and (iii) conservation of sequences present in already identified snoRNAs. Their importance in rRNA processing was demonstrated early on (for review see [2,3]), but their role in rRNA modification was not formally demonstrated until 1996, when box C/D snoRNAs were shown to be involved in 2′-O methylation ([4,5]), with box H/ACA snoRNAs subsequently implicated in pseudouridine formation [6,7,8].

Box H/ACA snoRNAs are involved in the base isomerization of uridine to generate pseudouridine (Figure 1A). They fold into two hairpin structures connected by a hinge containing the H motif (5′-ANANNA-3′), followed by a tail harboring the second ACA motif. The functional complex consists of a box H/ACA snoRNA specifying the site of modification on the rRNA, the core proteins Nhp2, Nop10, Gar1 and the Cbf5 pseudouridine synthase. Recognition of the substrate uridine is directed by two short sequences complementary to the rRNA and surrounding the target. These sequences are located on the opposite strands of the internal loops of the hairpins [7].

Ribose 2′-O methylations in rRNA (Nm) are driven by the box C/D snoRNA (Figure 1B). They possess highly conserved structural elements—the C/D and C′/D′ boxes—and sequences complementary to mature rRNA. The C (5′-RUGAUGA-3′)/D (5′-CUGA-3′) boxes establish partial base pairing and fold into a kink-turn (K-turn), facilitating interactions with proteins crucial for the modification process. The C′/D′ boxes are degenerate, but they also participate, through base pairing, to the overall single-hairpin structure of the box C/D snoRNA. The snoRNA is part of a small nucleolar ribonucleoprotein (snoRNP), which forms when Snu13 is recruited through specific binding with the K-turn [9], followed by the recruitment of Nop56, Nop58 and the methyltransferase Nop1. The nucleotide to be modified is determined by two elements: the interaction between the snoRNA and the rRNA through the formation of a Watson-Crick helix of 10 to 21 nucleotides (this length heterogeneity being due to heteroduplex thermodynamic stability or local pre-rRNA conformation) and the exact positioning of the modified nucleotide five nucleotides upstream from the D/D′ box, regardless of the nature of the base (A, G, U, C).

Interestingly, two orphan C/D snoRNAs—snR4 and snR45—were recently shown to be responsible for the acetylation of C_1773_ and C_1280_ in the 18S rRNA. The mechanism of action resembles that of the box H/ACA snoRNA, with the two guide sequences base pairing on either side of the target rRNA, generating a protruding bulge exposing the C substrate. This bulge is recognized by the Kre33 acetyltransferase, the only protein identified to date as associated with this snoRNP. These two snoRNAs have retained the ability to form a snoRNP with Nop1 and the other core proteins, but are no longer able to attach a methyl group to ribose, presumably because the base-pairing/distance rules are no longer respected [10].

In yeast, 30 box H/ACA snoRNAs are responsible for the pseudouridylation of 46 of the 47 sites in the four rRNA species. Some snoRNAs can direct the modification of up to four different positions within a single or different mature rRNA molecules. Similarly, 54 of the 55 2′-O methylations observed can be attributed to 43 box C/D snoRNAs. Most snoRNA genes exist as individual transcription units (TU), but a few are present in polycistronic TUs or are encoded by introns [11]. They undergo 3′ and 5′ processing and assemble with core proteins with the assistance of many other factors. These complex events are not dealt with here but have been reviewed elsewhere [12,13,14].

In addition to snoRNA-guided processes, several modifications are catalyzed by stand-alone enzymes. Most of these modifications are base modifications (mN), with the exception of the conversion of 5S-U_50_ into pseudouridine by Psu7 through recognition of the canonical UGΨAR sequence and 25S-G_2922_ 2′-O methylation, which is directed by Spb1. Surprisingly, Spb1 can also support 25S-Um_2921_ modification in the absence of snR52, making 25S-Um_2921_ the only site targeted by two independent modification processes [15].

*S. cerevisiae* displays six different types of base modification in the 18S and 25S rRNAs, at 10 sites: one N^7^-methylguanosine, two N^6^-dimethyladenosines, two N^1^-methyladenosines, two N^3^-methyluridines, two C^5^-methylcytosines and one complex N^1^-methyl-N^3^-aminocarboxypropylpseudouridine. All the enzymes responsible for these base modifications have been identified and their mode of action is known (Table 1).

## 3. SnoRNA Engineering

The mode of action of snoRNPs provides flexibility for the development of biotechnology tools. This has made it possible to modify rRNAs artificially for studies of the importance of individual nucleotides and the influence of the presence of a methyl group at positions that are not methylated in natural conditions.

Due to the architecture of the rDNA locus, with more than 200 copies of the rRNA genes, it is difficult to generate cellular ribosomes with modified rRNA. The very specific nature of the mechanism of snoRNA-guided modification (see above) has provided a tool for overcoming this obstacle. A study on U24, which methylates C_1436_ and A_1448_ in the 25S rRNA, showed that the deletion of a single nucleotide within the sequence complementary to the rRNA (the antisense element) shifted the D box one nucleotide downstream, resulting in the modification of U_1437_ instead of C_1436_ [16]. Modification of the antisense element from the mouse U20 similarly resulted in a new site-directed nucleotide modification [4], indicating that the modified nucleotide is specified exclusively by the D and D′ boxes and the antisense element. It has therefore been suggested that snoRNA engineering could be used to facilitate studies of the ribose methylation machinery, snoRNA function and the effect of point mutations within rRNA in vivo [17]. A plasmid based on the *snR38* gene expressed under the control of the *GAL1* promoter has been designed to target modification by simply replacing the antisense element with a specific sequence or with random sequences covering a large region of the rRNA (the 810 nt domain V, encompassing PTC). Studies with this plasmid have identified nucleotides essential for ribosome structure or function [18]. These random and specific approaches have been used for more detailed explorations of the interference effect on important nucleotides located within the PTC. The principal conclusion of these studies was that the observed defect was correlated with methylation itself, rather than the antisense effect. Most artificial modifications at positions unmodified in nature have no physiological consequences. Those leading to growth defects are also associated with slow protein synthesis, consistent with the interference effect observed for nucleotides already identified as involved in tRNA binding, subunit association or decoding [19]. Interestingly, the expression of a few engineered snoRNAs did not alter modification status. This lack of alteration was subsequently shown to be due to rRNA degradation, caused either by the modification itself, or by some unexplained snoRNP effect [20]. This finding raised questions about the need for snoRNA-rRNA co-evolution. A similar proof-of-concept was also established for the SnR36 H/ACA snoRNA, for which a substitution in the guide sequence led to an ectopic Ψ modification [6], but no tool has yet been developed for studying the ectopic targeting of Ψ within rRNA. This snoRNA-based tool was developed principally for rRNA modification, but it has also been successfully used to modify mRNA [21].

## 4. Detection of rRNA Modifications

A global view of rRNA modifications is required, to determine their roles. However, until recently and the development of NGS approaches, it was very difficult to develop such a view. Classical approaches for detecting modified ribonucleosides are based on thin-layer chromatography, capillary electrophoresis, or related techniques. In all cases, the sample is reduced to nucleotides, which are then separated on the basis of their chemical properties [22]. Unfortunately, these methods are labor-intensive, require the use of radioactive labeling and do not allow precise mapping of the modified residue. RP-HPLC, coupled or not with Mung bean protection assays, associated with tandem mass spectrometry methodologies can also be used [23,24] to quantify multiple RNA modifications accurately across conditions and cell types. However, although they are quantitative and reliable they require that the RNA of interest is isolated free from contaminants and partially digested, such methods cannot provide information about the molecule itself or the position of the modification. This technique is therefore suitable for rRNAs and tRNAs, which are abundant. Until the development of techniques based on deep sequencing (next-generation sequencing or NGS), it remained difficult to obtain a global view of the modifications carried by an RNA molecule at single-nucleotide resolution. NGS approaches have provided access to genome-wide quantitative data, shedding light on the positions only partially modified and those with a modification efficiency that varies as a function of the conditions tested. Various techniques are available for mapping RNA modifications [25,26,27] and chemical detection methods have been particularly successful in studies of rRNA modifications. One of these methods, RibomethSeq, is based on the protection, by ribose 2′-O-methylation, of the phosphodiester bond between nucleotides N and N + 1 generated by alkaline fragmentation. This protection is demonstrated by the absence of reads starting at the methylated position. This approach can be used to detect 2′-O-methylation in various types of RNA, including rRNA and tRNA [28,29,30,31]. Efficient quantitative mapping of pseudouridines by the HydraPsiSeq approach also recently became possible [26]. This method is based on RNA cleavage at random uridine residues by hydrazine, followed by aniline treatment for RNA chain scission at abasic sites. The protected residues (positions not cleaved by hydrazine) reveal the presence of pseudouridine residues. Conversely, m^7^G and m^3^C modifications can be detected on the basis of abasic site formation under alkaline conditions. These sites are then cleaved by aniline treatment and the 5′-phosphate generated is used for selective ligation for sequencing (Alkaniline-Seq). A frequency of m^7^G modifications as low as 2% can be detected with this method, which has been used to confirm the presence of a single m^7^G site, at position 1575, in the *S. cerevisiae* 18S rRNA (Table 1) [32]. These approaches are highly efficient, but modification-specific, because specific treatment of the RNA is required for detection of the modified positions.

Direct single-RNA molecule sequencing (without the need to generate cDNA) using the nanopore technology has the potential for the direct detection of all RNA modifications, provided that they disturb the resulting sequencing signal sufficiently, which is the case for the most common modifications: m^6^A, m^5^C, m^7^G and pseudouridines [33,34,35]. Direct RNA sequencing could also potentially be used to decipher the structure of single RNA molecules [36]. This approach is very promising but requires further improvement. In terms of bioinformatics pipelines, there is a real need to develop efficient scripts [37]. One major limitation is the need to sequence an artificial modification-free mirror RNA for comparison to the natural RNA [37]. The modified position is identified as a “sequencing error” at specific positions. As this approach is not hypothesis-dependent, the nature of the modification is completely unknown and further experiments are therefore required to determine the chemical nature of the modification.

## 5. Function in Translation Fidelity

The two most frequent rRNA modifications have a clear and significant effect on RNA structure. Indeed, the presence of multiple modified nucleotides influences the secondary and tertiary structure of the ribosome and its interactions with its partners: tRNA, mRNA and proteins. Pseudouridine provides an extra hydrogen bond that can take part in additional pairing interactions with RNA, thereby increasing backbone rigidity and the stability of the structure. It also enhances base stacking [38]. Hydrophobicity is increased by 2′-*O*-methylation, protecting against nucleolytic attack, stabilizing helices [39] and increasing steric hindrance. Based on recent analyses of the 2′-O-methylation of mRNA, it appears likely that Nm modifications also disrupt diverse other RNA interactions dependent on 2′-OH groups [40].

The mapping of modifications onto the three-dimensional (3D) structure of the ribosome highlights the non-random distribution of these modifications within the rRNA. The location of modifications depends on four criteria: (1) most modifications occur in conserved regions; (2) they are concentrated in the interior of the subunits and not exposed at the periphery of the ribosome, where protein concentration is high; (3) at least 60% of the modified nucleotides occur in functionally important regions; (4) the modifications are conserved throughout evolution (Figure 2) (for a review, see [41]). All of these observations suggest a possible effect of these modifications on translation. However, it is difficult to determine the importance of each single modification for translation fidelity, for many reasons.

First, almost all modifications occur very early on the 35S pre-rRNA transcript, through the association of the snoRNP or modification enzyme with the rRNA and are therefore involved in processing itself, ribosome assembly, export to the cytoplasm and quality control for the processed subunits. This is illustrated by Nop1 mutants, in which the methylation and assembly processes are uncoupled [42] (for a review on snoRNP function in ribosome biogenesis see [43]). Studies of the dual-function snR10, which is involved in rRNA processing and modification, have shown that the two functional domains (one for each function) act cooperatively [44].

Second, some snoRNAs target multiple positions (up to four, in both the 18S and 25S rRNAs, for snR49). Furthermore, some enzymes target other RNA species in addition to rRNA. This is the case for Kree33, which acetylates the leucine and serine tRNAs with the help of the Tan1 adaptor protein. Pus7 also has the ability to use multiple substrates: it catalyzes Ψ35 formation in the spliceosomal snRNA U2 [45] and pseudouridine conversion at position 13 of the cytosolic tRNA and position 5 of the pre-tRNA^tyr^ [46]. This site-directed pseudouridine synthase also mediates the modification of more than 200 uridine residues in mRNA upon heat-shock treatment [47], or about 260 mRNA sites during post-diauxic growth [48]. As mRNA and tRNA are the principal actors in translation, it remains difficult to determine the effect of rRNA modification in isolation.

The disruption of most individual snoRNAs, impairing single or multiple modifications, has been shown to have little or no impact on cell growth or translation, although some exceptions exist, such as 25S-Um_2921_ [49,50]. This particular modification site is highly conserved and under the control of two independent mechanisms dependent on a box C/D snoRNP and the stand-alone methyltransferase Spb. The abolition of this modification through coupled mutations of Spb1 and snR52 greatly decreases growth rates, alters polyribosome profiles and LSU structure and confers paromomycin sensitivity, whilst decreasing translational accuracy [15,51]. The combined loss of 25S-Gm_2288_ and 25S-m^5^C_2278_ also has a dramatic effect on 25S rRNA structure, impairing the recruitment of several ribosomal proteins.

Despite the rarity of examples of losses of individual modifications having a major impact on ribosome integrity and translation efficiency, a growing number of studies have concluded that the modification state of the ribosome has an impact on the decoding capacity of the ribosome. The first evidence in support of this conclusion was provided by treatment with ethionine, which inhibits methylation and by the *nop1.3* mutant allele of the methyltransferase Nop1, which abolishes methylation without disturbing pre-rRNA processing other than by slightly delaying late 60S rRNA processing. Both these conditions result in a severe impairment of growth, indicating that methylation is important for ribosome structure, or directly required for translational accuracy [42]. This hypothesis was confirmed by the description of the D95A mutant allele of *Cbf5A*, which abolishes Ψ formation and results in a low growth rate [52] and poor translational accuracy [53] with no major defect of rRNA processing. Conversely, single box C/D snoRNA deletions individually subjected to functional profiling for growth or antibiotic susceptibility revealed subtle effects under nonoptimal growth conditions [54]. The analysis of combinatorial snRNA deletions targeting modifications of the same functional region of the ribosome has proved a major advance. Dissections of the peptidyl transferase center, the inter-subunit bridges and the decoding center suggested that modifications acted together to fine-tune translation and adapt the ribosomes to functional requirements. The influence of modifications to peptide bond formation and decoding was investigated in strains lacking up to six pseudouridine residues, including the aforementioned snR10-guided Ψ_2919_. Synergic effects on growth rate, protein synthesis and ribosome structure were observed [55], together with effects on translational accuracy [51]. This detrimental effect was attributed to structural changes affecting tRNA accommodation at the A site. Inter-subunit bridges have a key function, as they promote associations between subunits and movements from one subunit relative to the other, thereby controlling the various steps of translation. Two such bridges, the B2a (helix 69 of the LSU) and B1a (helix H38 of the SSU, also known as the A-site finger) bridges, have been studied, as they contain numerous modifications. H69 in the 25S rRNA and the SSU H44 together form the B2a bridge, establishing close contact with the A-site and P-site tRNAs. A role for this bridge in translation fidelity has already been demonstrated [56]. Five modifications, four Ψ and one 2′-O methylation, conserved in humans, are present in this small domain. The Fournier laboratory showed, in standard combinatory studies, that it was the loss of modifications, rather than the impairment of snoRNP-rRNA association, that enhanced rRNA turnover [57]. As expected, the loss of one or two modifications led to changes in elongation and termination accuracy, highlighting the key role of this region. Surprisingly, increasing the number of snoRNA deletions had the opposite effect, with antibiotic resistance observed [58]. Similarly, the LSU H38, known as ASF, constitutes the B1a inter-subunit bridge, contacting the A-site tRNA via the tip of the stem-loop, which is devoid of modifications. It is very rich in Ψ, with six positions in H38 and another four positions in the two adjacent helices. Deletion analysis showed that yeast cells are less sensitive to the loss of modifications in H38, the three central Ψ in the loop (Ψs 1004, 1042 and 1052) being more important for subunit association [59]. Translational accuracy is barely affected, with a very specific effect on UGA readthrough in the absence of the six modifications [58]. The decoding center has also been studied. This center has the particular feature of consisting of rRNA segments separated in the primary sequence, but coming together in the tertiary structure to surround the E, A and P-sites of the ribosome. Eight modifications are present (5 Nm and 3 Ψ, including the highly modified m^1^acp^3^Ψ_1191_), seven of which are conserved in humans. Remarkably, m^1^acp^3^Ψ is not conserved in *E. coli* but corresponds to the modified m^2^G_966_ which contributes to the initiation step of translation (PMID: 23530111, 22649054). These modifications may come into contact with either the tRNA o0r the mRNA, thereby influencing the decoding capacity of the ribosome. Indeed, it has been concluded that modifications located above the A-site tRNA (Cm_1428_ and Ψ_1187_) and in the P-site tRNA (Cm_1639_ and m^1^acp^3^Ψ_1191_) influence the rates of growth and translation, antibiotic resistance [60], the termination process and reading frame maintenance [58]. Finally, a methylation cluster is present in helices H70 and H71 of domain IV in the 25S rRNA. This region harbors 3 2′O-methylation and one 5-methyl cytosine, which is of particular interest (see below). Competition assays showed that the absence of Cm_2288_ and/or m^5^C_2278_ affected the fitness of yeast cells and ribosome stability [61]. These experiments, all performed before 2014, suggest that rRNA nucleotide modifications are important to ensure a stable structural conformation of the ribosome, but may, to a lesser extent, modulate the decoding capacity of the ribosome, depending on the particular combination of modifications present. Unfortunately, at the time at which these studies were performed, technology had not yet advanced sufficiently to obtain a global view of the modification status of the ribosomes within the cell population, to confirm the possibility of a diverse composition capable of adapting gene expression to environmental changes.

## 6. Specialized Ribosomes

Previous studies have demonstrated a link between the particular effects of certain modifications on precise aspects of translational accuracy and the heterogeneous nature of ribosome modification patterns. It is, therefore, of particular interest to determine whether this could influence translation of specific mRNAs. This issue was first addressed in yeast, in studies of lifespan and stress resistance. In a screen aiming to identify lifespan mediators, the methyltransferase Rcm1 was shown to be downregulated in chronologically aged cells. This enzyme drives 25S-m^5^C_2278_ cytosine methylation. These cellular processes are accompanied by a decrease in general protein synthesis, raising questions about the involvement of rRNA modification in aging. Indeed, *RCM1* deletion completely abolish C_2278_ methylation, resulting in a longer lifespan and greater stress resistance after exposure to H_2_O_2_. An investigation of the pattern of mRNA recruitment to polysomes showed that the absence of m^5^C_2278_ results in a completely different set of mRNAs being translated relative to that translated by methylated ribosomes; more surprisingly, this set of mRNAs resembled the mRNAs of wild-type cells exposed to oxidative stress (Figure 3). The authors of the study concerned concluded that cells depleted of m^5^C_2278_ were in a pre-active state of responsiveness to translational stress that might account for their longer lifespan and greater stress resistance [62]. It remains unclear whether *RCM1* activity is actually modulated in response to cellular signals. Epigenetic inactivation of the human counterpart of *RCM1*, *NSUN5*, has been observed in 38% of glioma-derived cell lines displaying the expected decreases in protein synthesis and, more surprisingly, a reprogramming towards stress proteins, enabling the tumor cells to cope with hostile environments [63]. The impact of a second stand-alone methyltransferase has also been investigated. Rrp8 directs the m^1^A_645_ modification in domain I of the 25S rRNA [64]. As expected, the loss of this modification affects the local structure of the 60S subunit, disturbing interactions with ribosomal proteins, especially eL32. -accordingly, polysome profiling shows halfmers typical of 43S initiation complexes awaiting the addition of the 60S subunit. The authors concluded that this observation reflected alterations to the recruitment of the 60S subunit, rather than an initiation defect *per se*. They explored the consequences of this defect for cellular mRNA translation, by performing two-dimensional gel analysis (DIGE) and mRNA determinations for proteins displaying changes in expression. Overall, 18 of the 1900 proteins detected displayed >1.5-fold differences in expression and clustered either with enzymes involved in carbon metabolism (with Sol3 protein regulated exclusively at the protein level) or with proteins related to translation itself. The authors concluded that this modification affected the translation of a subset of mRNAs with metabolic roles, either directly, or indirectly, due to the impact on components of the translational machinery [65]. In wild-type cells cultured in standard conditions, the nucleotide displays almost 100% modification. No conditions altering this modification have ever been found, raising questions about the potential regulation of this modification. However, certain diseases, such as ribosomopathies, may be partly explained by the impact of the lost modifications on the production on particular proteins. The concept of specialized ribosomes, differing in terms of modification status, was recently illustrated by a study of the rare dimethyl adenosine modification [66]. This modification, driven by the essential dimethylase Dim1 [67], concerns exclusively the two adjacent adenosines, A_1781_ and A_1782_, within the 18S rRNA. These residues are located in the decoding center and are universally conserved. The authors demonstrated the coexistence of a monomethylated m^6^A form within translating ribosomes, also under the control of Dim1, albeit present at only very low levels (4%). Interestingly, the monomethylation-to-dimethylation ratio changes specifically in conditions of sulfur deprivation (10% m^6^A in the polysomal fraction). This sensing of sulfur availability is mediated by Dim1 itself. Similar observations have been reported for cultured mammalian cells, indicating that this adaptation to sulfur deprivation is conserved among eukaryotes. The impact of the switch from m^6^_2_A to m^6^A on mRNA translation has been investigated by ribosome profiling, a powerful method providing quantitative and qualitative information on translation at whole-cell level (for a review, see [68,69]). Using the D87E Dim1 mutant, which can tolerate 80% m^6^A modification with no effect on rRNA processing or subunit formation, the authors showed a significant and specific effect on the translation efficiency (TE) of 16 genes relative to WT (4% m^6^A); 12 of these genes are involved in sulfur metabolism. As expected, this differential behavior disappeared in conditions of sulfur deprivation. Based on these findings, the authors suggested that the modification status of the ribosome conferred specificity for decoding certain mRNAs closely linked to environmental signals. However, the molecular mechanism underlying this specificity remains unknown and ribosomal heterogeneity may also reflect the presence of different ribosomal proteins (Figure 3).

The notion of specialized ribosomes is not limited to yeast or unicellular organisms and seems to be generalizable to all organisms. In multicellular organisms, cells are generally less subject to environmental stresses. Conversely, in these organisms, protein synthesis must be adapted and synchronized to different tissues and developmental phases. For these reasons, it may be beneficial to have different populations of ribosomes, for the fine regulation and synchronization of translation according to the needs of the body.

In human cancer cells, which have high proliferation rates, a hyperactivation of ribosome biogenesis occurs, leading to high levels of protein synthesis. This is made possible, in part, by the deregulation of RNA polymerase I-mediated rRNA transcription due to a loss of p53 function [70]. Fibrillarin (FBL), the mammalian homolog of Nop1 and snoRNAs are also overexpressed in breast cancer [71]. FBL expression is also downregulated by p53 and p53 inactivation therefore leads to increases in fibrillarin activity and, thus, rRNA hypermethylation [72]. This modification pattern, in turn, alters the fidelity of translation, with more efficient IRES-dependent translation initiation for the growth factor IGF-1R. An FBL knockdown experiment was conducted to provide an additional demonstration of the link between rRNA methylation and the specificity of translation for particular mRNAs [73]. As expected, ribosome biogenesis was affected, but only mildly, as inactivation was achieved with an siRNA under controlled transfection conditions. Methylation status was investigated by ribomethseq, which revealed site-specific variations rather than a global effect, with the identification of partially methylated sites potentially subject to regulation. It was, therefore, possible to explore the specificity of such “variant” ribosomes by ribosome profiling. Translation was found to be up- or downregulated for a small set of mRNAs, 8% of which contained IRES. This category of mRNAs is enriched in genes implicated in tumorigenesis: oncogenes (p53, c-myc), growth factor receptors (IGF-1R, VGEF, FGF) or apoptosis modulators. Thus, in the context of cancer cells, fibrillarin levels and specific patterns of rRNA modification can modulate translation, thereby contributing to tumorigenesis (Figure 3).

Progress has been made with studies of methylation status based on ribomethseq on adult and embryonic tissues from mice. It was first established that the sites targeted for methylation in mouse rRNA have an equivalent in HeLa and HTC116 human cancer cell lines [31]. Conversely, in searches for sites modified in humans but not in the tissues of adult mice, the authors investigated the methylation status of A_1310_ and the presence in the genome of SNORD126, the corresponding guide snoRNA, in both mice and rats. They confirmed the absence of Am _1310_ in mice, but found that this modification was tissue-specific in rats, due to the differential expression of SNORD126. The physiological consequences of this unexpected observation were not explored further. One third of human sites are partially methylated, but the vast majority of sites from the adult mouse tissues studied have methylation rates above 90%. Using this important observation as a starting point, the authors were able to detect decreases in methylation level. Indeed, embryonic tissues (E9.5 and E16.5) are characterized by a set of 59 sites with lower levels of methylation in at least one tissue. This elegant work then focused on Gm_4593_, which displays a change in methylation rate from 63% in E9.5 to 0% in adult brain tissue. The corresponding snoRNA, SNORD78, is hosted by intron 6 of the *GAS5* gene, which contains eight other snoRNAs distributed between the other 10 introns. The authors showed that SNORD78 was no longer present in adult brain cells, a situation resembling that for rat SNORD126, except that the other eight SNORDs present within the RNA molecule were not affected. This discrepancy can be attributed to the alternative splicing of exon 7 in adult cells, resulting in defective SNOD78 processing. The biological purpose of this ribosome heterogeneity remains unclear, but this work highlights the modulation of rRNA methylation during developmental process (Figure 3).

## 7. Perspectives

During the last decade, the progress in our knowledge of rRNA modifications has been overwhelming. Despite the lack of ribosome structures at a sufficiently high resolution for the direct visualization of modifications in many cases, advances in our ability to map and quantify precisely major rRNA modifications have proved a real breakthrough, leading to the emergence of the concept of “specialized ribosomes”. This notion of “specialized ribosomes” is sometimes difficult to impose on the human context, due to difficulties demonstrating the specificity of ribosome action, but also because, in most cases, studies are performed in the context of disease. Thanks to the co-evolution of guide snoRNAs and ribosomes, *S. cerevisiae* constitutes an ideal model for addressing these questions more efficiently, in a much simpler system than human cells. Indeed, it remains difficult to perform systematic position-by-position studies of each modification in humans, but approaches of this kind are entirely feasible in yeast. Moreover, as yeasts are subject to highly fluctuating environments, it would be interesting to study the relationship between the pattern of modification and the response to environmental stress, which would undoubtedly provide important clues to the role of ribosome modifications in humans.

## Figures and Tables

**Figure 1 ijms-22-07419-f001:**
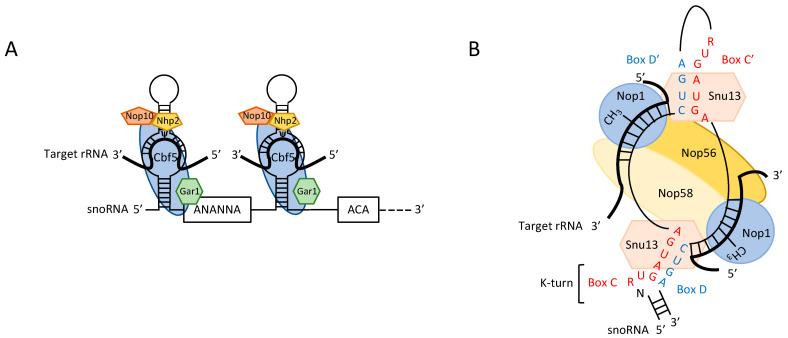
Structures of box H/ACA and box C/D snoRNP. (**A**) Diagram of a typical box H/ACA snoRNP. The target rRNA is in bold and is bound to complementary regions of the H/ACA guide snoRNA. The uridine residues converted into pseudouridine lie within a bulge known as the pseudouridine pocket and are indicated (Ψ). The two H and ACA sequences are boxed. Associated proteins are shown in color. (**B**) Diagram of a typical box C/D snoRNP. Two target rRNA regions are represented in bold and are bound to complementary regions of the C/D guide snoRNA. The nucleosides methylated lie 5 nucleotides upstream from a D or D′ box and are indicated (-CH_3_). The C and C′ boxes are highlighted in red and the D and D′ boxes are highlighted in blue. The associated proteins are shown in color.

**Figure 2 ijms-22-07419-f002:**
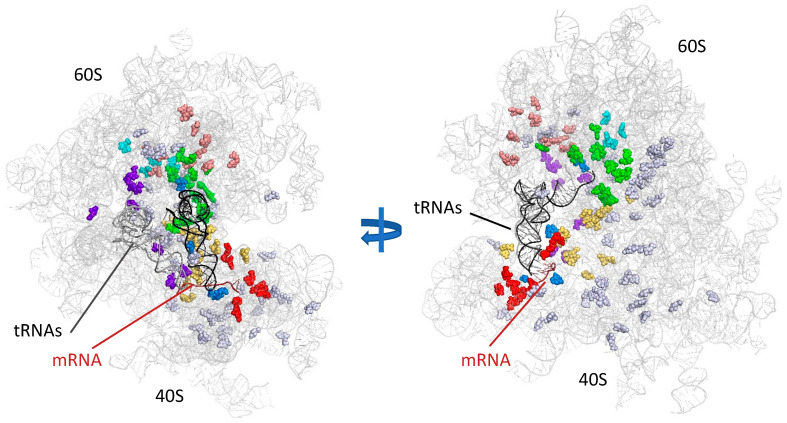
Localization of rRNA modifications on the ribosome. The modified positions are indicated by spheres on the structure of the yeast ribosome (PDB: 1VXX). The P-site tRNA is indicated in black, the E-site tRNA is shown in dark gray and the mRNA is indicated in dark red. Each modified nucleotide is shown in color according to its position with respect to the functional regions of the ribosome (green: PTC; red: A-site; cyan: peptide-exit channel; blue: P-site; purple: E-site, pink: A-site finger; yellow: inter-subunit bridges). Positions lying outside the functional regions are shown in gray. Blue arrow represents a rotation by 90°.

**Figure 3 ijms-22-07419-f003:**
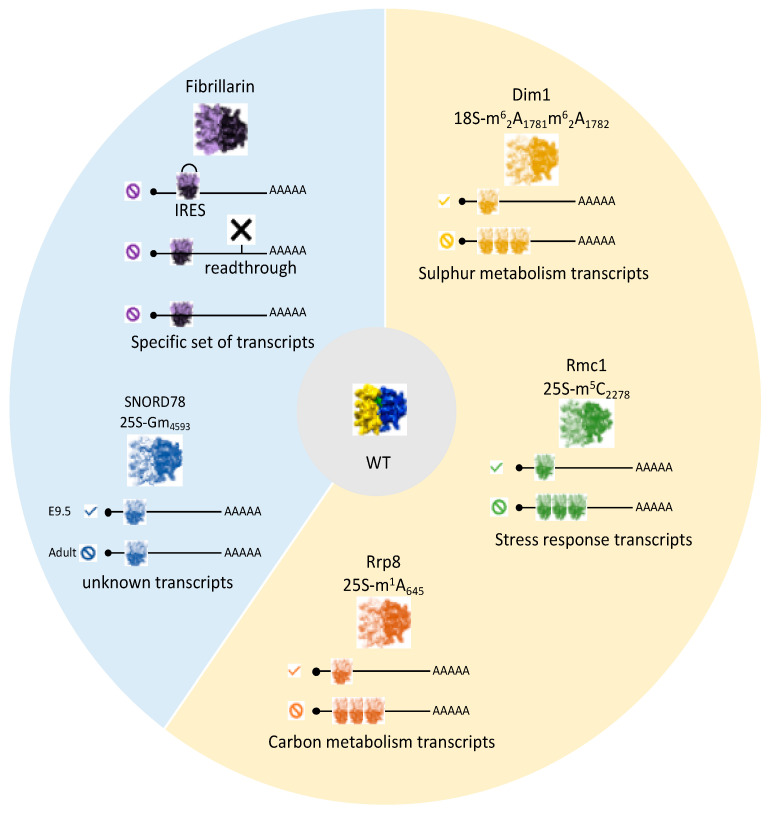
Models of specialized ribosomes in yeast and mammalian cells. The yellow part of the diagram illustrates examples related to yeast and blue part illustrates those related to human or mouse cells. (🛇) absence or (✓) presence of the modification concerned.

**Table 1 ijms-22-07419-t001:** Listing of the rRNA modifications present in *Saccharomyces cerevisiae*. The modification type, position and the snoRNA or enzyme responsible are given, together with the modification rate, conservation between species (H for Human and Ec for *Echerichia coli*) and location within functional or structural domains of the ribosome.

Modification	Position	Enzyme/snoRNA	Modification Rate	Conservation	Functional Domain
Ψ	5S-50	Pus7	>85%	H	
Ψ	5,8S-73	snR43	<85%		
Am	18S-28	snR74	>85%	H	
Am	18S-100	snR51	<85%	H	
Ψ	18S-106	snR44	>85%	H	
Ψ	18S-120	snR49	>85%	H	
Ψ	18S-211	snR49	<85%		
Ψ	18S-302	snR49	>85%		
Cm	18S-414	U14	>85%	H	
Am	18S-420	snR52	>85%	H	
Am	18S-436	snR87	>85% d	H	
Ψ	18S-466	snR189	<85%		
Am	18S-541	snR41	>85%	H	
Gm	18S-562	snR40	>85% d		
Um	18S-578	snR77	>85%	H	DC-A site
Am	18S-619	snR47	>85%	H	
Ψ	18S-632	snR161	<85%	H	
Ψ	18S-759	snR80	>85%	H	
Ψ	18S-766	snR161	>85%	H	
Am	18S-796	snR53	>85%		
Am	18S-974	snR54	>85%	H	
Ψ	18S-999	snR31	<85%	H	IS bridge DC-E site
Cm	18S-1007	snR79	>85%		IS bridge DC-E site
Gm	18S-1126	snR41	>85%		IS bridge
Ψ	18S-1181	snR85	>85%	H	IS bridge
Ψ	18S-1187	snR36	>85%	H	DC-A site
m^1^acp^3^Ψ	18S-1191	snR35,Emg1, Tsr3	>85%	H	DC-P site
Um	18S-1269	snR55	>85%	H	DC-A site
Gm	18S-1271	snR40	>85%	H	DC-A site
ac^4^C	18S-1280	snR4, Kre33	>85%	H	DC-A site
Ψ	18S-1290	snR83	>85%	H	
Ψ	18S-1415	snR83	<85%		
Gm	18S-1428	snR56	>85%	H	DC-A site
Gm	18S-1572	snR57	>85%		E-site
m^7^G	18S-1575	Bud23	>85% d	H	
Cm	18S-1639	snR70	>85% d	Ec, H	DC-P site
ac^4^C	18S-1773	snR45, Kre33	>85%	H	IS bridge
m_2_^6^A	18S-1781	Dim1	>85%	Ec, H	IS bridge
m_2_^6^A	18S-1782	Dim1	>85%	Ec, H	IS bridge
m^1^A	25S-645	Bmt1 (Rrp8)	>85%	H	PTC
Am	25S-649	U18	>85%	H	Peptide exit tunnel
Cm	25S-650	U18	>85%		Peptide exit tunnel
Cm	25S-663	snR58	<85% d	H	Peptide exit tunnel
Ψ	25S-776	snR80	>85%		
Gm	25S-805	snR39b	>85%	H	Peptide exit tunnel
Am	25S-807	snR39, snR59	>85%	H	
Am	25S-817	snR60	>85%	H	
Gm	25S-867	snR50	>85% d		
Am	25S-876	snR72	>85% d		
Um	25S-898	snR40	>85% d		
Gm	25S-908	snR60	>85%	H	
Ψ	25S-960	snR8	>85%	H	A site finger helix 37
Ψ	25S-966	snR43	>85%	H	A site finger helix 37
Ψ	25S-986	snR8	>85%		A site finger helix 38
Ψ	25S-990	snR49	>85%		A site finger helix 38
Ψ	25S-1004	snR5	<85%	H	A site finger helix 38
Ψ	25S-1042	snR33	>85%	H	A site finger helix 38
Ψ	25S-1052	snR81	>85%	H	A site finger helix 38
Ψ	25S-1056	snR44	>85%		A site finger helix 38
Ψ	25S-1110	snR82	>85%		
Ψ	25S-1124	snR5	>85%	H	A site finger helix 39
Am	25S-1133	snR61	>85%	H	A site finger helix 39
Cm	25S-1437	U24	>85%	H	Peptide exit tunnel
Am	25S-1449	U24	>85% d	H	
Gm	25S-1450	U24	>85%	H	
Um	25S-1888	snR62	>85%	H	
Ψ	25S-2129	snR5	>85%		
Ψ	25S-2133	snR3	>85%	H	
m^1^A	25S-2142	Bmt2	>85%		Subunit surface
Ψ	25S-2191	snR32	>85%	H	
Cm	25S-2197	snR76	>85% d	H	E-site
Am	25S-2220	snR47	>85%	H	E-site
Am	25S-2256	snR63	>85%	H	IS bridge helix 69 A site tRNA
Ψ	25S-2258	snR191	>85%	Ec, H	IS bridge helix 69 A site tRNA
Ψ	25S-2260	snR191	>85%	Ec, H	IS bridge helix 69
Ψ	25S-2264	snR3	>85%	H	IS bridge helix 69
Ψ	25S-2266	snR84	>85%	H	IS bridge helix 69 P-site tRNA
m^5^C	25S-2278	Bmt3 (Rcm1)	>85%	H	IS bridge
Am	25S-2280	snR13	>85%		IS bridge
Am	25S-2281	snR13	>85%	H	IS bridge
Gm	25S-2288	snR75	>85%	H	IS bridge
Ψ	25S-2314	snR86	>85%	H	
Cm	25S-2337	snR64	>85%	H	
Ψ	25S-2340	snR9	>85%	H	
Um, Ψ, Ψm	25S-2347	snR65, snR9		H	
Ψ	25S-2349	snR82	>85%	H	
Ψ	25S-2351	snR82	>85%		
Ψ	25S-2416	snR11	>85%	H	
Um	25S-2417	snR66	>85%		
Um	25S-2421	snR78	>85%	H	E-site
Gm	25S-2619	snR67	>85%	Ec, H	P-site
m^3^U	25S-2634	Bmt5	>85%		PTC
Am	25S-2640	snR68	>85%		
Um	25S-2724	snR67	>85%		
Um	25S-2729	snR51	>85% d	H	
Ψ	25S-2735	snR189	>85%	H	
Gm	25S-2791	snR48	>85%		E-site
Gm	25S-2793	snR48	>85%	H	E-site
Gm	25S-2815	snR38	>85%	H	PTC
Ψ	25S-2826	snR34	>85%	H	PTC
m^3^U	25S-2843	Bmt6	>85%	H	
Ψ	25S-2865	snR46	>85%	H	PTC
m^5^C	25S-2870	Bmt4 (Nop2)	>85%	Ec, H	PTC
Ψ	25S-2880	snR34	>85%	H	PTC
Um	25S-2921	snR52, Spb1	>85%	Ec, H	PTC
Gm	25S-2922	Spb1	>85%	H	PTC
Ψ	25S-2923	snR10	>85%	H	PTC
Ψ	25S-2944	snR37	<85%	H	PTC
Am	25S-2946	snR71	>85%	H	PTC
Cm	25S-2948	snR69	>85% d		PTC
Cm	25S-2959	snR73	>85%	H	PTC
Ψ	25S-2975	snR42	<85%	H	PTC

d: discrepancy, with different values in different publications.

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
