# Peer review of "Saccharomyces cerevisiae, a Powerful Model for Studying rRNA Modifications and Their Effects on Translation Fidelity"

_ijms, 2021, doi:10.3390/ijms22147419_

Round 1
Reviewer 1 Report
In the review article " Saccharomyces cerevisiae, a Powerful Model for Studying rRNA Modifications and Their Effects on Translation Fidelity" the authors review the different modifications present in rRNA and their impact on ribosome function. This is a comprehensive well written review that I recommend for publication.
Some minor comment:
Line 25, the authors should specify that the rRNA nomenclature refers to yeast and not eukaryotes (in mammals the 25S is 28s rRNA, and the 35S is 42S).
Line 159, please change snRNAs with snoRNAs.
Line 205, the authors should specify that the direct single-RNA molecule sequencing is performed with the Nanopore system.
Line 414, there is an extra “f” before FBL
Reviewer 2 Report
A review “Saccharomyces cerevisiae, a Powerful Model for Studying rRNA 2 Modifications and Their Effects on Translation Fidelity” highlights advancements in our understanding of the rRNA modification functional role. Authors focus the review on several issues related to the rRNA modification in yeast as a convenient model organism. The role of snoRNAs and methods of modified residues detection are discussed in much details, as well as an influence of modifications on translation fidelity and a concept of specialized ribosomes.
The review is informative, up to date and interesting to follow. In general, it is of interest to the community.
There are several minor issues that have to be resolved.
- line 31: duplication “with with”
- line 34: duplication “positions have been shown to be modified positions”
- Table 1 contains an entry for m3U956 25S rRNA modification. In the ” 3D RIBOSOMAL MODIFICATION MAPS DATABASE” this position is reported as m5U following cautious suggestion of Bakin et al., Biochemistry (1994) 33:13475-83. Later this was disproved in Sharma et al., Nucleic Acids Res. (2014) 42:3246-60 and not included into a catalogue of Yang et al., PLoS ONE (2016) 11:e0168873.
- The chapter 4 “Detection of rRNA modifications” is focused mainly on NGS based methods, while mass-spectrometry based methods and not necessarily coupled to Mung bean nuclease treatment and RP-HPLC, but also MALDI MS based are more confident. There are much more methods which are more quantitative and reliable (although less throughput).
- line 296 “t-RNA” should be replaced with”tRNA”
- line 321 Statement “m1acp3Y also conserved in E. coli” is not correct. The corresponding E. coli nucleotide m2G966 is also modified although not identical to human m1acp3Y.
- line 414 “ f FBL” should probably be replaced with “FBL”.
